



# Spatial and temporal variability of solar penetration depths in the Bay of Bengal and its impact on SST during the summer monsoon

Jack Giddings[1], Karen J. Heywood[1], Adrian J. Matthews[2], Manoj M. Joshi[1], Benjamin G. M. Webber[1], Alejandra Sanchez-Franks[3], Brian A. King[3] and Puthenveettil N. Vinayachandran[4]

[1]Centre for Ocean and Atmospheric Sciences, School of Environmental Sciences, University of East Anglia, Norwich, NR4 7TJ, UK.
[2]Centre for Ocean and Atmospheric Sciences, School of Environmental Sciences and School of Mathematics, University of East Anglia, Norwich, NR4 7TJ, UK.
[3]National Oceanography Centre, Southampton, SO14 3ZH, UK.
[4]Center for Atmospheric and Oceanic Sciences, Indian Institute of Science, Bangalore, India.

*Correspondence to*: Jack Giddings (j.giddings@uea.ac.uk)

**Abstract.** Chlorophyll influences regional climate through its effect on solar radiation absorption and thus sea surface temperature (SST). In the Bay of Bengal, the effect of chlorophyll on SST has been demonstrated to have a significant impact on the Indian summer (southwest) monsoon. However, little is known about the drivers and impacts of chlorophyll variability in the Bay of Bengal during the southwest monsoon. Here we use observations of downwelling irradiance measured by an ocean glider and three profiling floats to determine the spatial and temporal variability of solar absorption across the southern Bay of Bengal during the 2016 summer monsoon. A two-band exponential solar absorption scheme is fitted to vertical profiles of photosynthetically active radiation to determine the effective scale depth of blue light. Scale depths of blue light are found to vary from 12 m during the highest (0.3–0.5 mg m$^{-3}$) mixed layer chlorophyll concentrations, to over 25 m when the mixed layer chlorophyll concentrations are below 0.1 mg m$^{-3}$. The Southwest Monsoon Current and coastal regions of the Bay of Bengal are observed to have higher mixed layer chlorophyll concentrations and shallower solar penetration depths than other regions of the southern Bay of Bengal. Substantial sub-daily variability in solar radiadion absorption is observed, which highlights the importance of near-surface ocean processes in modulating mixed layer chlorophyll. Simulations using a one-dimensional K-profile parameterisation ocean mixed layer model with observed surface forcing from July 2016 show that a 0.3 mg m$^{-3}$ increase in chlorophyll concentration increases sea surface temperature by 0.35°C in one month with SST differences growing rapidly during calm and sunny conditions. This has the potential to influence monsoon rainfall around the Bay of Bengal and its intraseasonal variability.

## 1. Introduction

Absorption of incoming solar radiation at the ocean surface modulates the upper ocean heat content, which controls the exchange of heat and moisture to the lower troposphere (Zaneveld et al., 1981; Lewis et al., 1990). Water containing chlorophyll absorbs more solar irradiance than clear water, modifying the vertical heating profile of the upper ocean and thus



sea surface temperature (SST; Morel, 1988; Morel and Antoine, 1994; Ohlmann, 2003), which in turn can affect the large-scale ocean circulation and climate (Sweeney et al., 2005; Wetzel et al., 2006).

The concentration of chlorophyll in the Indian Ocean has been shown to have a significant effect on the South Asian monsoon (Nakamoto et al., 2000; Wetzel et al., 2006; Turner et al., 2012; Giddings et al., 2020). Imposing seasonally varying chlorophyll concentrations in the Bay of Bengal (BoB) has been found to increase SST by 0.5°C, which can increase rainfall by up to 3 mm day$^{-1}$ over Myanmar during the South Asian summer (southwest) monsoon onset and over northeast India and Bangladesh during the Autumn intermonsoon period (Giddings et al., 2020). In the Arabian Sea, the inclusion of seasonally

varying chlorophyll due to phytoplankton blooms in a coupled climate model led to a 50% reduction in mixed layer depth biases, an increase in local SST, and a subsequent increase in rainfall of up to 2 mm day$^{-1}$ over western India during the southwest monsoon onset (Turner et al., 2012). Chlorophyll concentrations of 0.3 mg m$^{-3}$ in the Arabian Sea during boreal spring have been found to increase SST by 0.4°C compared with simulations using globally constant attenuation rates corresponding to near-zero chlorophyll concentrations in an ocean isopycnal general circulation model (GCM; Nakamoto et

al., 2000). Coupling a biogeochemistry model to a coupled ocean-atmosphere GCM to derive chlorophyll-dependent attenuation rates of solar radiation led to an SST increase of 1°C in the Arabian Sea during autumn, and an increase in summer monsoon rainfall of 3 mm day$^{-1}$ along the west coast of India (Wetzel et al., 2006). However, little is known about the influence of surface chlorophyll on the temporal and spatial variability of solar penetration depths across the BoB and how surface chlorophyll directly impacts on SST during the summer southwest monsoon.

Remote sensing of chlorophyll pigments from satellites has demonstrated that chlorophyll concentrations vary substantially in both space and time, suggesting a corresponding spatial and temporal variability of solar penetration depth (Nakamoto et al., 2000; Murtugudde et al., 2002). Chlorophyll-dependent optical parameters such as the diffuse attenuation coefficient ($K_d$), defined as the fraction of solar radiation attenuated per unit distance through the upper ocean, can be determined from in situ radiometer measurements (e.g., Smith and Baker, 1981) and estimated using ocean colour data from satellites (e.g., Lee et al.,

55    2005).

Previous studies have parameterised solar penetration depths as a function of remotely sensed chlorophyll concentration for certain solar absorption schemes. Morel and Antoine (1994) produced high-order polynomial relationships for a two-band model that related the scale depths of blue and red light (300–750 nm) to surface chlorophyll concentration, assuming an idealised Gaussian vertical profile of chlorophyll to a depth of one solar penetration depth. Ohlmann (2003) used the

HYDROLIGHT radiative transfer model (Ohlmann and Siegel, 2000) to produce vertical profiles of solar radiation for pre-defined chlorophyll concentrations, time of day and cloud cover to determine optical parameters. A scale depth relationship was developed for the transmission of the ultraviolet-visible spectrum (300–750 nm) as part of a two-band model. However, there remains uncertainty over which of these parameterisations is most applicable for use in a specific region or climate model.

The oceanic components of state-of-the-art GCMs have various chlorophyll-dependent parameterizations and associated

solar absorption schemes. For example, the Community Earth System Model (CESM; Kay et al., 2015) uses the Ohlmann (2003) chlorophyll-dependent parameterisation for a two-band model (Smith et al., 2010). Meanwhile, the Institute Pierre





Simon Laplace climate model (IPSL; Mignot et al., 2013) uses a 3-band model where light is split into red, green and blue wavebands that each have a chlorophyll-dependent attenuation coefficient (Lengaigne et al., 2007). Both GCMs have the capability to assimilate satellite ocean colour measurements. Ocean colour measurements have revolutionized our understanding of how chlorophyll-induced heating affects ocean dynamics and the climate system (Murtugudde et al., 2002; Sweeney et al., 2005; Wetzel et al., 2006). However, the limited spatial and temporal resolution of these GCMs and assimilated ocean colour data mean they inadequately resolve mesoscale and sub-seasonal chlorophyll concentration variability, which might influence the intraseasonal variability of BoB SST and the South Asian summer monsoon.

Across the southern BoB, the seasonal reversal of wind direction during the boreal summer creates conditions conducive for chlorophyll blooms. Southwesterly monsoon winds initiate the Southwest Monsoon Current (SMC), which flows northeastward, advecting cooler, saline water from the Arabian Sea and the western equatorial Indian Ocean around the southernmost point of India and Sri Lanka into the warmer and fresher BoB (Fig. 1b; Jensen, 2003; Sanchez-Franks et al., 2019). The SMC evolves into a shallow, narrow and fast-moving current with surface speeds of up to 0.6 m s$^{-1}$ and a thickness of up to 550 m (Webber et al., 2018). Large chlorophyll blooms along the southwestern coastal shelf of India, initiated by upwelling nutrients, become entrained in the SMC and are advected around the south of Sri Lanka into the central BoB in summer (Lévy et al., 2007). A tongue of high surface chlorophyll concentrations extends into the central BoB, following the path of the SMC (Fig. 1a). The bloom is sustained east of Sri Lanka in the cyclonic eddy of the Sri Lanka Dome (SLD), identified as a region of lower absolute dynamic topography and cyclonic current vectors in Fig. 1b. Open-ocean Ekman upwelling in the SLD brings nutrients to the near-surface to support the phytoplankton population (Vinayachandran and Yamagata, 1998; Vinayachandran et al., 2004; Thushara et al., 2019). Hence, the high surface chlorophyll concentrations associated with the SMC and SLD are expected to lead to reduced solar penetration depths throughout the summer monsoon period.

The large freshwater flux from river output and rainfall in the BoB during boreal summer creates a barrier layer, where strong salinity stratification forms within the isothermal layer and below the mixed layer (Vinayachandran et al., 2002; Sengupta et al., 2016). The presence of the barrier layer isolates the mixed layer above from cooling by entrainment (Duncan and Han, 2009). Instead, the surface heat flux forcing, such as shortwave radiation and turbulent heat fluxes, primarily controls the warming and cooling phases of the surface ocean (Li et al., 2017). The barrier layer has been found to influence BoB SST (Drushka et al., 2014) and its thickness impacts the summer monsoon intraseasonal oscillation (Li et al., 2017). The additional effects of localised biological heating from surface chlorophyll could amplify the warming in these shallow mixed layers. Understanding the mesoscale and sub-seasonal solar penetration depth variability and its impact on BoB surface ocean properties would highlight the direct effect of chlorophyll concentration at finer spatial and temporal scales.

In this study, we determine (i) how solar penetration depth varies temporally and spatially across the southern BoB; (ii) how near-surface chlorophyll concentrations affect solar penetration depths; (iii) how chlorophyll concentration directly impacts on SST in the southern BoB. To quantify the influence of chlorophyll on solar penetration depth and SST, an ocean glider and three profiling floats were deployed as part of the joint India-UK Bay of Bengal Boundary Layer Experiment


(BoBBLE; Vinayachandran et al., 2018) to measure in situ physical, optical and biogeochemical variables in the upper ocean during July 2016 at high horizontal and temporal resolution. We fit a two-band solar absorption function to observed vertical profiles of photosynthetically active radiation (PAR). PAR is an integral of downwelling irradiance between 400 to 700 nm (blue to red light), allowing us to determine the solar penetration depth.

An overview of the data and methods is presented in Section 2. Section 3 presents an analysis of the temporal and spatial variability of determined $h_2$ (Section 3.1), and a comparison of determined $h_2$ and observed chlorophyll concentration to two previously published parameterisations (Section 3.2). This is then followed by an analysis of five idealised simulations with an imposed solar penetration depth from the $h_2$ observations to investigate the impact of observed chlorophyll on upper ocean radiant heating rate and SST in the southern BoB. The simulations were conducted using the one-dimensional K-profile
parameterisation ocean mixed layer model (Section 3.3). Section 4 presents the discussion and conclusions.

## 2. Data and methods

### 2.1 Observations and instruments

*a. Ocean gliders and Argo profiling floats*

During the BoBBLE field campaign (Vinayachandran et al., 2018), a Seaglider (SG579) was deployed at 86° E on 30 June
2016 along a transect at 8° N east of Sri Lanka and piloted to 85.3° E by 8 July, where the glider continued to take measurements until 29 July 2016. The glider profiled on a sawtooth trajectory from the surface to 700–1000 m, completing a full dive cycle approximately every 4 hours. The glider was equipped with a Seabird Electronics (SBE) conductivity (salinity), temperature and depth (CTD) sensor, a Wetlabs Triplet Ecopuck measuring chlorophyll-a fluorescence and optical backscatter at wavelengths 470 nm and 700 nm and a Biospherical Instruments quantum scalar irradiance PAR ($\mu$E m$^{-2}$ s$^{-1}$) sensor measuring
visible wavelengths between 400 nm and 700 nm. The Wetlabs and PAR sensors sampled to a depth of 300 m with a vertical resolution of ~1 m. Quality control was performed on the entire conductivity–temperature (CT) dataset using Conservative Temperature–Absolute Salinity (IOC et al., 2010) space analysis and further quality control in depth space for individual vertical profiles. Salinity spikes were removed when the glider vertical speed was less than 0.035 m s$^{-1}$ as the unpumped CT sensor relied on a suitable flow of water for reliable measurements. The ocean glider PAR measurements were factory
calibrated. The CT sensor was factory calibrated and was then further calibrated against in situ ship CTD observations.

Argo profiling floats 629, 631 and 630 that are part of the international Argo float program were deployed along the 8° N transect at 85.5° E, 87° E and 89° E on the 28 June, 1 July and 4 July respectively, where they sampled to 500 m daily until mid-August and every other day until the end of September. All three floats were equipped with SBE 41N CTD and a Satlantic OCR-504 ICSW radiometer measuring downwelling irradiance at wavelengths 380 nm, 490 nm, 555 nm ($\mu$W cm$^{-2}$ nm$^{-1}$) and
PAR ($\mu$E m$^{-2}$ s$^{-1}$). The CTD measurements were factory calibrated and radiometer measurements were factory calibrated with channel-specific coefficients. The vertical resolution on the ascent to the surface was ~1 m for the radiometer and CTD.



### b. PAR

All night-time PAR profiles (local solar zenith angles greater than 70°) and low-light PAR profiles (maximum PAR in the
top 5 m is less than 100 $\mu$E m$^{-2}$ s$^{-1}$) measured by the glider and profiling floats were discarded. Further quality control was
carried out to remove the effect of external environmental factors, which include shading by passing clouds that causes sudden
fluctuations in light intensity, and wave-focusing that creates sawtooth spiking in the vertical PAR profiles (Zaneveld et al.,
2001). All PAR data between 0–5 m depth were removed from the analysis, as noise from wave-focusing obscured the signal
of the absorption of solar radiation. A quality-control method using a fourth-degree polynomial, modified from the
methodology of Organelli et al. (2016), was used to identify PAR perturbations below the near-surface and remove profiles
displaying excessive noise.

### c. Chlorophyll

The glider's raw fluorescence voltages were converted into chlorophyll-a concentrations according to the manufacturer
calibrations. Since phytoplankton that are exposed to too much sunlight trigger the non-photochemical quenching mechanism
to protect themselves from photooxidative damage (Müller et al., 2001), chlorophyll-a fluorescence is suppressed near the
surface in the daytime. To correct for quenching, nighttime fluorescence-to-backscatter ratios were used to derive corrected
daytime chlorophyll-a fluorescence profiles (Thomalla et al., 2018). The glider fluorescence-derived chlorophyll-a
concentrations, after correcting for non-photochemical quenching, showed values that were higher than those derived from the
shipboard CTD chlorophyll-a fluorescence sensor. Concentrations were calibrated by applying a scale factor and offset derived
using linear regression between the glider and CTD chlorophyll-a profiles.

The profiling floats did not make chlorophyll-a fluorescence measurements, so a novel approach was developed to derive
chlorophyll-a concentration from radiometer data alone (see Appendix A for method details). Chlorophyll-a strongly absorbs
light at 490 nm wavelength so the vertical gradient of $E_d(490)$, the downwelling radiation flux at 490 nm, was used to derive
a proxy for in situ chlorophyll-a concentration to identify the vertical distribution of chlorophyll-a. Vertical profiles of the
natural log of $E_d(490)$ were individually corrected for their mean in situ dark count calculated from measurements below 200
m (Organelli et al., 2016). Profiles displaying excessive noise were eliminated using the fourth-degree polynomial method of
Organelli et al. (2016). The attenuation coefficient $K_d$ was calculated for each 1 m discretised layer. The attenuation coefficient
$K_d$ is the sum of the attenuation of pure seawater ($K_w$), represented as a constant, and the attenuation due to biology ($K_{bio}$), a
chlorophyll-a component (Morel and Maritorena, 2001; Xing et al., 2011). Further quality control is applied to vertical profiles
of $K_{bio}$ before chlorophyll-a is calculated using empirically determined coefficients from Morel et al. (2007) (Fig. A1). The
chlorophyll-a pigment concentration that was derived from radiometry data or remotely sensed by satellite will henceforth be
referred to as "chlorophyll" for convenience.






*d. Satellite products*

The remotely sensed chlorophyll concentrations used in this study are sourced from the European Space Agency's Ocean Colour - Climate Change Initiative (ESA OC-CCI; Lavender et al., 2015) version 3.1 (available at http://www.esa-oceancolour-cci.org). The OC-CCI project involved the merging of remotely sensed chlorophyll concentrations from MODIS, MERIS, SeaWiFS and VIIRS radiance sensors to provide a continuous dataset ranging from 1997–2016 with increased spatial coverage of the global oceans. The radiance sensors on the satellites detect the water-leaving radiance at specific wavelengths to estimate chlorophyll concentration. The thickness of ocean surface layer "seen" by the radiance sensors is approximately one solar penetration depth or the depth where downwelling irradiance decreases to $1/e$ of the surface irradiance (Gordon and McCluney, 1975), depending on the local chlorophyll concentrations. 8-daily and monthly composites of chlorophyll concentration with spatial resolutions of 4 km have been used to investigate the weekly and monthly variability of chlorophyll concentration influencing solar penetration depths in the deployment region across the southern BoB from July to September 2016.

Satellite-derived absolute geostrophic velocities (meridional and zonal components) and absolute dynamic topography are altimeter products produced by SSALTO/Duacs, distributed by AVISO (https://www.aviso.altimetry.fr) and are available through the Copernicus Marine Environment Monitoring Service (http://marine.copernicus.eu). The daily composites of absolute geostrophic velocities and absolute dynamic topography have a spatial resolution of 0.25° x 0.25° and are used to investigate the surface current velocities that control chlorophyll concentration advection from July to September 2016.

**2.2 Methods**

The penetration depth of solar radiation is frequency dependent, with higher frequencies ("blue" light) having a much larger penetration depth than lower frequencies ("red" light). A two-wavelength double-exponential function (Paulson and Simpson, 1977) can be used to parameterise this behaviour in ocean models (e.g., Sweeney et al., 2005) and in mixed-layer heat budget studies (e.g., Vialard et al., 2008; Girishkumar et al., 2017). For this study we use a double-exponential function of the form:

$$Q(z) = q_2 \left[ \left( \frac{R}{1-R} \right) e^{-\frac{z}{h_1}} + e^{-\frac{z}{h_2}} \right] + d \qquad (1)$$

where $Q$ (W m$^{-2}$) is the irradiance at depth $z$ (m). Surface irradiance just below the ocean surface for blue light is denoted by $q_2$ (W m$^{-2}$). The scale depths, $h_1$ and $h_2$, represent the e-folding depths (m) of absorption of red and blue light respectively. The parameter $R$ is the ratio of the flux of red light to the total and is a measure of the partition of the solar flux into the arbitrary red and blue bands. An offset $d$ (W m$^{-2}$) has also been introduced to allow for a non-zero instrument response at zero radiation flux, when applied to a radiometer. In practice, $d$ is very small (compared with $q_2$).

Paulson and Simpson (1977) determined the optical parameters ($R$, $h_1$ and $h_2$) for each of the five Jerlov water types, which represent the range of turbidity observed in open ocean water (Jerlov, 1968). Water type I represents low open ocean chlorophyll concentrations of 0 to 0.01 mg m$^{-3}$ (Morel, 1988), where $h_1$ and $h_2$ are 0.35 m and 23 m respectively. Water type III represents high open ocean chlorophyll concentrations of 1.5 to 2.0 mg m$^{-3}$ (Morel, 1988), where $h_1$ and $h_2$ are 1.4 m and





7.9 m respectively. The penetration depth of blue light ($h_2 \sim 20$ m) is much larger than that for red light ($h_1 \sim 1$ m), and hence variations in $h_2$ exert the main control on the radiant heating of the surface ocean mixed layer and thus SST.

Optical parameter $R$ is 0.58 in water type I and 0.78 in water type III. The two-band model is an arbitrary approximation of the full solar spectrum, and there is no *a priori* definition of the value of the cut-off frequency between the red and blue bands. Hence, the parameter $R$ is allowed to vary, along with $h_2$, to maximise the fit of the two-band model to the data. The
variation of $R$ should not be interpreted as a physical change in the fraction of red light at the surface, which of course is independent of the ocean conditions below. Instead, the variation of $R$ should be interpreted as a degree of freedom in fitting a simple two-band scheme to model the full solar spectrum.

In this study we use Eq. (1) to fit to profiles of PAR reported in terms of moles of photons ($\mu$E m$^{-2}$ s$^{-1}$) instead of units of energy (W m$^{-2}$). For PAR measurements, the conversion of units from $\mu$E m$^{-2}$ s$^{-1}$ to W m$^{-2}$ can only be an approximation as
the PAR instrument measures photons across a range of visible wavelengths, but the exact spectrum across that range is unknown at any particular time (Sager and McFarlane, 1997). Although the absolute values of PAR change with unit conversion, the attenuation rate of visible light with depth and thus the value of $h_2$ is independent of the unit conversion of PAR. Hence, in practice we fit Eq. (1) to profiles of PAR with units of $\mu$E m$^{-2}$ s$^{-1}$ to determine values of $h_2$ and avoid PAR conversion uncertainty.

From the excessively noisy 1-m vertical resolution PAR measurements close to the surface we are unable to determine the transmission of red light (values of $R$ and $h_1$). We assume Jerlov water type IB (Paulson and Simpson, 1977) to be applicable to our region, based upon initial determinations of $h_2 \sim$17 m from fitting Eq. (1). We therefore constrain $R$ to be 0.67 and $h_1$ to be 1 m and thus fit PAR profiles between 5 m and 100 m to the transmission of blue light with depth ($h_2$) using Eq. (1) (Fig. 2a). The same fit plotted in log space (Fig. 2b) results in a near straight line below 5 m, demonstrating that the decrease in
PAR can be approximately represented with a single exponential below this depth. The contribution of the fixed parameters used for the fit was estimated by varying $R$ and $h_1$ between Jerlov water type I to III from Paulson and Simpson (1977) and varying the depth of removed near-surface PAR between 3–7 m. We combine uncertainties associated with the varied parameters and associated with fitting, to produce the overall uncertainty in the derived values of $h_2$.

**3. Results**

**3.1 Glider and profiling float observations**

The SLD is a prominent feature in the southwest BoB during the summer monsoon and is typically associated with high surface chlorophyll concentrations (Thushara et al., 2019). At the start of July 2016, the SLD is centred around 85–86° E and 5–10° N to the west of the SMC (Fig. 1b). Glider SG579 is located inside the SLD from 30 June and observes the weakening
of this cyclonic eddy after 2 July, remaining in a localised region between 85–86° E (Fig. 1c; black diamond). The average mixed layer salinity and temperature are 34 g kg$^{-1}$ and 28°C respectively (Fig. 3a and 3b). Chlorophyll concentrations peak on 1 July with values of 0.8 mg m$^{-3}$ at a depth of 18 m, indicating high surface chlorophyll concentrations (Fig. 3d). Corresponding



values of $h_2$ decrease from an average of 16 m on 30 June to 13 m on 1 July, as the average 0–30 m chlorophyll concentration increases from 0.2 mg m$^{-3}$ to 0.5 mg m$^{-3}$ in one day (Fig. 3d; black circles).

After 2 July, the SLD weakens and shifts towards the northwest, but the SMC continues to flow into the south-central BoB. Patches of surface chlorophyll, with concentrations of 0.1–0.4 mg m$^{-3}$ (Fig. 1d), continue to be advected by the SMC into the glider SG579 deployment region (85–86° E) until 19 July. Within the SMC, the mixed layer warms to 29°C and freshens to 33.3 g kg$^{-1}$ (Fig. 3a and 3b). Chlorophyll concentrations below the mixed layer remain around 0.5 mg m$^{-3}$ forming a deep chlorophyll maximum between 30–50 m depth (Fig. 3d). Meanwhile average 0–30 m chlorophyll concentrations decrease to

less than 0.2 mg m$^{-3}$ (Fig. 3d) and the corresponding average values of $h_2$ increase to more than 20 m until 16 July (Fig. 4a; dashed black line). The position and velocity of the SMC relative to the biologically productive southern coast of Sri Lanka determines how much surface chlorophyll is entrained and advected into the south-central BoB (Vinayachandran et al., 2004). Throughout most of July the SMC is too far south to intercept the high surface chlorophyll concentrations along the southern coast of Sri Lanka (Fig. 1d), explaining why in situ surface chlorophyll concentrations are relatively low after 2 July (Fig. 3d).

The variability of $h_2$ in the SMC is large (Fig. 4a). Values ranged between 15–31 m from 4 July onwards, which we partly attribute to sub-daily temporal variability in the mixed layer and surface chlorophyll concentrations. However, the derived $h_2$ values from glider SG579 are associated with relatively high uncertainty (typically ±2 m) due to the fitting of the double exponential function to noisy vertical PAR profiles, which may contribute to this apparent variability.

    The profiling float dataset allows us to extend the glider dataset temporally and spatially, providing daily measurements of

solar penetration depths until mid-August and then measurements every 2 days until the end of September, spanning much of the southern BoB. The vertical profiles of downwelling irradiance measured from the profiling floats are less noisy than those measured from the glider. Hence, the profiling floats display lower uncertainty in determined values of $h_2$ when compared with the glider (Fig. 4; a–d). As the SMC flows northeastward into the south-central BoB during early July, the surface current bifurcates. The main branch flows northward towards 10° N and the smaller branch flows eastward towards 90° E (Fig. 1d).

Fig. 5b shows the longitudinal variations of $h_2$ across the SLD and SMC. Values of $h_2$ decrease as remotely sensed chlorophyll concentrations increase towards the centre of the SMC (Fig. 5a and 5b), consistent with previous studies that show the SMC increasing chlorophyll concentrations in the region (e.g., Vinayachandran et al., 2004; Thushara et al., 2019). Float 631 is deployed on the eastern flank of the SMC and completes an anticyclonic loop, intercepting the eastern flank of the SMC a second time on 20 July at 87° E (Fig. 1d). Between 20–24 July the time series shows the mixed layer cooling, increasing in

salinity and deepening to 40 m depth, as barrier layer thickness increases to 40 m (Fig. 6a and 6b). Surface chlorophyll concentrations are patchy as the float intercepts the SMC with average mixed layer chlorophyll concentrations varying daily between 0.1–0.4 mg m$^{-3}$ (Fig. 6d). Average values of $h_2$ are around 16 m, varying between 10 to 20 m, similar to the sub-daily variability of $h_2$ observed from the glider in the SMC.

    Observations on the western side of the basin from 8–11° N show average $h_2$ values of 20 m compared with the average $h_2$

values of 16 m in the SMC (Fig. 5a). The timeseries of chlorophyll concentration from the westernmost float 629 shows the





mixed layer depth increasing from 25 m to below 50 m and the deep chlorophyll maximum deepening from 30 m to 50 m between 16 July to 13 August (Fig. 7d). Away from the SLD and SMC float 629 encounters a more transparent upper ocean with increased $h_2$ and reduced mixed layer chlorophyll concentration of 0.2–0.3 mg m$^{-3}$. Closer to the East India continental shelf, the influence of the freshwater runoff from rivers entering the basin enhances the supply of biological material and the

nutrient supply to the upper water column (Lotliker et al., 2016). Sedimentary material also reduces the solar penetrative depths and increases solar absorption in the surface layers of the coastal region. As a result, $h_2$ is reduced to the west of 83°E (Fig. 5b), associated with higher remotely sensed chlorophyll concentrations in this region (Fig. 5a). On 13 September, surface geostrophic velocities from satellite altimetry show an anticyclonic eddy moving eastward away from the East India coast (not shown) intercepting the path of float 629, causing the mixed layer to shoal and salinity to increase by 0.6 g kg$^{-1}$ in two days

(Fig. 7b). Average 0–30 m chlorophyll concentrations increase to 0.4 mg m$^{-3}$ and corresponding $h_2$ values decrease to a minimum of 11 m (Fig. 7d).

Daily variations in salinity of 0.2 g kg$^{-1}$ are observed by float 630 during 6–12 July, with the highest salinity recorded at 34.4 g kg$^{-1}$ in the mixed layer and the barrier layer on 10 July (Fig. 8b), possibly due to eddies shearing off from the main SMC flow (Fig. 1d). Values of $h_2$ are around 16 m as average chlorophyll concentrations of ~0.2 mg m$^{-3}$ in the surface 0–30 m (Fig.

8d) are entrained by the SMC and advected into the path of float 630 at around 89° E in early July. Towards the end of September, the SMC influence at 89° E reduces and the current shifts to the western side of the basin (Fig. 1f), consistent with climatological observations (Webber et al., 2018). Consequently, at 89° E a southeastward flow containing water from the eastern side of the basin along with some recirculated surface water from the SMC is observed (Fig. 1e and 1f). Float 631 yields $h_2$ values greater than 20 m (Fig. 6d), possibly indicating that the southeastward flow advects low surface chlorophyll

concentrations from the biologically unproductive eastern side of the BoB. We hypothesise that the displacement of the SMC to the western BoB would lead to reduced solar penetration depth in the west and increased solar penetration depth in the east during the summer.

### 3.2 Relationship between scale depth and chlorophyll concentration

Visible radiation in the upper ocean decreases by approximately 63% $(1 - e^{-1})$ from the surface to a depth equal to one scale depth. Glider observations show that over 80% of PAR is absorbed to a depth of 30 m (Fig. 3c). The chlorophyll concentration of the surface layer, where the majority of visible radiation is absorbed, is a key control on the amount of visible radiation absorbed and thus on the radiant heating rate of the surface layer. We examine the relationship between the average chlorophyll concentration in the surface layer and $h_2$, both observed by the glider. The average mixed layer depth in the glider time series

(Fig. 3d) and the determined maximum $h_2$ is approximately 30 m. Hence, we calculate the average chlorophyll concentration in the surface layer between 0 and 30 m depth. We do not derive a relationship between chlorophyll and $h_2$ from the profiling floats, since the float chlorophyll concentration is itself derived from vertical profiles of light absorption ($E_d(490)$).

As expected, $h_2$ is inversely related to chlorophyll concentration (Fig. 9). Observed average chlorophyll concentrations from glider SG579 vary by a factor of 6 during the BoBBLE campaign. Larger $h_2$ values of ~20 m are associated with lower



mixed layer chlorophyll concentrations of less than 0.3 mg m$^{-3}$; smaller $h_2$ values of ~12 m are associated with higher mixed layer chlorophyll concentrations of 0.35 mg m$^{-3}$.

The observations compare well with two commonly used double exponential parameterisations in ocean GCMs relating light absorption to chlorophyll concentration (Fig. 9; Table 1), from Morel and Antoine (1994) [MA94] and Ohlmann (2003) [O03]. We assume for the O03 two-band solar absorption scheme that the incident angle of solar radiation on the ocean surface

and the cloud index are both zero. Both the parameterisations define a power law dependence in scale depth as a function of chlorophyll, with the greatest change in scale depth occurring at lower chlorophyll concentrations, between 0.08–0.1 mg m$^{-3}$, and the smallest change in scale depth occurring at higher chlorophyll concentrations above 0.2 mg m$^{-3}$ (Fig. 9). The determination coefficients ($r^2$) of O03 and MA94 against the observations show that these functions fit similarly to determined values of $h_2$. The parameterisations predict scale depths to be within ±3.6 m of the determined $h_2$. For chlorophyll

concentrations larger than 0.2 mg m$^{-3}$ MA94 and O03 predict scale depths smaller than the determined $h_2$, although the number of observations above this concentration is limited. From our results, we cannot definitively select the most appropriate parameterisation given the spread and uncertainty in the $h_2$ estimates.

### 3.3 Implications of chlorophyll concentration on BoB SST

The determined values of $h_2$ for each glider and float timeseries varies by a factor of two (Fig. 4; e–h). The 5$^{th}$ and 95$^{th}$ percentile of all $h_2$ values are 14 m and 26 m respectively. With the majority of solar radiation absorbed in the surface mixed layer, then the difference between $h_2 = 14$ m and $h_2 = 26$ m would have significant effects on the radiant heating of the surface layer and SST. We can compare the impact these two values of $h_2$ would have on the temperature change for an idealised water column. The temperature change is related to the daily average solar radiant heating rate of a layer of upper ocean with

thickness, $H$, as

$$\frac{dT}{dt}\Big|_Q = \frac{Q_{sw}(0) - Q_{sw}(H)}{\rho c_p H} = \frac{Q_{sw}(0) - (1-R)Q_{sw}(0)e^{-\frac{z}{h_2}}}{\rho c_p H} \qquad (2)$$

where we specify $H = 30$ m to represent the average mixed layer depth from the glider, $\rho = 1021$ kg m$^{-3}$ to represent the average density of seawater in the upper 30 m from the glider dataset and $c_p = 3850$ J kg$^{-1}$ K$^{-1}$ to represent the specific heat capacity of sea water. The daily average solar irradiance absorbed in this mixed layer is calculated by taking the difference between the

daily average solar irradiance incident on the ocean surface, $Q_{sw}(0)$, and daily average solar irradiance at the base of the mixed layer, $Q_{sw}(H)$. At depths greater than 5 m, we assume all red light is absorbed and $Q_{sw}(H)$ is then the blue light radiation flux that penetrates the base of the mixed layer.

The daily average solar irradiance incident on the column surface is estimated to be 280 W m$^{-2}$ based on solar irradiance measurements during clear sky conditions during the observation period (Vinayachandran et al., 2018). For the purposes of

this calculation, we ignore the effects of advection, entrainment and mixing, as well as any atmospheric feedbacks from changing SST (Vijith et al., 2020). The average determined value of $h_2$ for July, August and September is indicative of Jerlov water type IB where $h_2 = 17$ m (Fig. 4e–4h), hence we use a constant value of $R = 0.67$ for the same Jerlov water type. If the




water column has an $h_2$ value of 26 m, then the solar irradiance absorbed in the upper 30 m would be 251 W m$^{-2}$ with 29 W m$^{-2}$ absorbed below 30 m. If the water column has an $h_2$ value of 14 m then the solar irradiance absorbed in the upper 30 m would

be 269 W m$^{-2}$ with 11 W m$^{-2}$ absorbed below 30 m. Using Eq. (2) the increased absorption of solar irradiance in the mixed layer when $h_2$ decreases from 26 m to 14 m leads to a 0.35°C month$^{-1}$ increase in radiant heating rate, confirming that chlorophyll-induced heating over the determined range of $h_2$ will lead to significantly different values of SST.

These idealised calculations are now extended to investigate further the influence of near-surface chlorophyll concentrations on SST and heat distribution of the upper ocean. A one-dimensional K-profile parameterisation (KPP) model

(Large et al., 1994) is used to run five idealised simulations with five constant $h_2$ values of 14 m, 17 m, 19 m, 21 m and 26 m, which represent the 5th, 25th, 50th, 75th and 95th percentile of all determined values of $h_2$ respectively throughout July 2016. The model has a simple two-band solar radiation scheme, identical to Paulson and Simpson (1977), to replicate the transmission of solar radiation in the upper ocean. Initial KPP sensitivity experiments, not presented in this paper, show that the influence of $R$ on SST is not negligible. Hence, five constant values of $R$ from Paulson and Simpson (1977) are chosen with $R = 0.58$ when

$h_2 = 26$ m (Jerlov water type I), $R = 0.62$ when $h_2 = 21$ m and 19 m (Jerlov water type IA), $R = 0.67$ when $h_2 = 17$ m (Jerlov water type IB) and $R = 0.77$ when $h_2 = 14$ m (Jerlov water type II). The influence of $h_1$ on SST is negligible and is fixed at 1 m (Jerlov water type IB) for each of the five idealised simulations. The model mixed layer depth is defined as the depth where the bulk Richardson number is equal to a critical value of 0.3 (Large et al., 1994). Horizontal advection, Ekman pumping and atmospheric feedbacks are absent from the model by design.

The mean vertical profiles of temperature and salinity from the glider for 1–10 July provide the subsurface (0–1000 m depth) initial conditions. Hourly solar shortwave flux is derived from the downwelling shortwave radiation observed every 2 minutes from the RAMA (Research Moored Array for African-Asian-Australian Monsoon Analysis and Prediction; McPhaden et al., 2009) mooring at 8° N, 90° E in the southern BoB approximately 4° east of the glider location. The hourly rainfall data are interpolated from three-hourly rainfall rate from the Tropical Rainfall Measuring Mission (TRMM; Huffman et al., 2007)

for the same location. The sensible and latent heat fluxes and the surface wind stress are sourced from TropFlux (Kumar et al., 2012) at a daily resolution, which are then linearly interpolated to an hourly resolution. TropFlux is used as it provides an accurate representation of heat fluxes during the boreal summer in the BoB (Sanchez-Franks et al., 2018). Evaporation rates are calculated from the latent heat flux from TropFlux at the same hourly resolution. The model is spun up for one month using the surface forcing data for June 2016. For this spin up period, the scale depth of blue light was fixed at the Jerlov water type

IB value of $h_2 = 17$ m. After the spin up, the model was run through July 2016 in five configurations with $h_2$ equal to 14 m, 17 m, 19 m, 21 m and 26 m.

The BoBBLE campaign took place during a suppressed period of convection or a break phase in the South Asian monsoon. The South Asian monsoon is subject to active-break cycles on subseasonal timescales (10 to 30 days) driven by the Boreal Summer Intraseasonal Oscillation (BSISO; Wang and Xie, 1997), which are strongly influenced by air-sea interactions

(Sengupta et al., 2001). Associated with this break phase, no precipitation is recorded, and solar shortwave flux remains high during the campaign between 4–15 July (Fig. 10b and 10c), allowing for strong diurnal heating of the ocean surface during





this period. By 15 July, precipitation increases (Fig. 10c) as deep atmospheric convection enters the campaign region marking the transition into an active phase of the BSISO.

The KPP experiments demonstrate that changing $h_2$ from 26 m to 14 m leads to an increase in daily average SST of 0.35°C by the end of July 2016 (black line; Fig. 10e). The average mixed layer depth is 34 m and remains relatively constant during July. Hence, the previous idealised calculation was a good approximation as we estimated a similar amount of radiant heating for a mixed layer of comparable thickness. Decreasing $h_2$ from 26 m to 17 m, 19 m and 21 m, leads to progressively smaller increases in daily average SST from 0.25°C, 0.18°C and 0.14°C by the end of July 2016, respectively (Fig. 10e). The maximum diurnal change in SST for the $h_2 = 14$ m simulation is 1.0°C, compared with 0.62°C for the $h_2 = 26$ m simulation (Fig. 10d). From 1–15 July the SST from the $h_2 = 14$ m simulation warms at the greatest rate of 0.04°C day$^{-1}$, compared with 0.02°C day$^{-1}$ for the $h_2 = 26$ m simulation (Fig. 10d). From 15 July onwards, during an active phase of the BSISO, SST warming for the $h_2 = 14$ m simulation is just 0.01°C day$^{-1}$, compared with the slight SST cooling in the $h_2 = 26$ m simulation (Fig. 10d). Decreased solar penetration depth leads to increased absorption of solar radiation over a shallower depth of ocean. Hence, the mixed layer warms and the water below the mixed layer cools as less solar radiation penetrates deeper in the water column (Fig. 10h).

On the 25 July, high precipitation rates of 4 mm day$^{-1}$ freshen the ocean surface (Fig. 10g), which contributes to an increase in mixed layer salinity stratification and a reduction in the maximum mixed layer depth in all five simulations (Fig. 10f). A reduction in wind stress also partly contributes to the reduction in the maximum mixed layer depth, as wind-driven turbulent mixing is reduced (Fig. 10c). The mixed layer in the $h_2 = 26$ m simulation shoals to a maximum depth of 30 m and recovers to a previous depth of 34 m a day later (Fig. 10f). Conversely, the mixed layer in the $h_2 = 14$ m simulation shoals to a maximum depth of 23 m and recovers to a previous depth of 34 m five days later (Fig. 10f). Decreased solar penetration depth and increased solar radiation absorption further increase mixed layer thermal stratification and stability, which amplifies and prolongs the vertical and temporal change in mixed layer depth. Shoaling the mixed layer to a depth comparable to the solar penetration depth increases the sensitivity of SST to changes in chlorophyll concentration (Turner et al., 2012; Giddings et al., 2020). Hence, freshwater input through precipitation and additional biological warming through the occurrence of high chlorophyll concentrations in the SMC and SLD region would enhance SST increase during an active BSISO phase, which would potentially have a positive impact on atmospheric convection.

## 4. Discussion and clonclusions

Observed and inferred chlorophyll concentrations show a deep chlorophyll maximum at 50 to 80 m across the Southern BoB during the southwest monsoon, with higher near-surface chlorophyll concentrations occurring intermittently within the SMC, SLD and coastal regions. The average $h_2$ for July, August and September is indicative of Jerlov water type IB ($h_2 = 17$ m). The $h_2$ values display temporal and spatial variability on sub-daily timescales, a consequence of sub-daily variability of surface chlorophyll concentrations entrained by the SMC. In the SLD and SMC, where high surface chlorophyll concentrations





are advected into the southern BoB, $h_2$ is generally lower. The bifurcation of the SMC, and hence of the chlorophyll entrained

in its flow, reduces $h_2$ values to the south and east of the SMC as filaments and eddies break off from the main current. Away

from the SMC, the upper ocean is more transparent with $h_2$ values of more than 20 m. In coastal regions, $h_2$ values occasionally

reduce to 11 m due to high surface chlorophyll concentrations, as well as other chlorophyll pigments, detritus material and

other biological constituents.

This study has shown that gliders and floats are suitable oceanographic platforms to determine $h_2$ from observed PAR

profiles. PAR profiles measured by the glider tended to be noisier than those measured by the floats. The removal of

excessively noisy PAR measurements in the top 5 m means the fit of Eq. (1) to PAR profiles is not constrained at the surface,

so the optical parameters of red light are not constrained. Instead, fixed water type IB values of $R$ and $h_1$ are used to replicate

red light absorption in the top 5 m of all PAR profiles, with minimal influence on the determined values of $h_2$ below 5 m. This

demonstrates that solar penetration depths for blue light can be determined from in-water PAR profiles measured from gliders

and floats without near-surface PAR measurements and without the need for labour-intensive and costly ship-based tethers,

CTD rosettes, and buoys used in previous studies (e.g., Ohlmann et al., 1998; Lotliker et al., 2016).

The O03 and MA94 scale depth parameterisations demonstrate similar correlation and RMSE when compared with

determined $h_2$ values and average 0–30 m chlorophyll concentrations. Both parameterisations demonstrate a power law

relationship of scale depth as a function of chlorophyll and predict scale depths to be within ±3.6 m of the determined $h_2$,

although both tend to underestimate $h_2$ for chlorophyll concentrations of 0.2–0.5 mg m$^{-3}$. The spread and uncertainty of the

determined $h_2$ means we cannot robustly select the most appropriate parameterisation to predict scale depth in this region.

The relationship between determined $h_2$ and observed chlorophyll concentrations measured from the glider has limitations.

Determined $h_2$ not only represent the attenuation of blue light due to chlorophyll-a concentration, but also the attenuation of

blue light due to other biological constituents and other suspended particles. Furthermore, the observed chlorophyll

concentration is only a proxy for actual chlorophyll-a concentration. Hence, determined $h_2$ values potentially overestimate blue

light attenuation due to chlorophyll-a pigments, affecting the relationship between determined $h_2$ and average observed

chlorophyll concentration and the fit of MA94 and O03 shown in Section 3.2. Future climate modelling studies should consider

different types and concentrations of biological constituents that affect $h_2$, such as coloured dissolved organic matter (e.g., Kim

et al., 2018).

Relatively low blue-light scale depths are likely to occur within the SMC and SLD due to the higher surface chlorophyll

concentrations that will in turn lead to locally enhanced warming. The width of the SMC is approximately 300 km (Webber et

al., 2018) and surface chlorophyll begins to increase in April and typically peaks in July (Lévy et al., 2007) resulting in a

considerable area and duration of enhanced biological surface warming. Likewise, the eastern and western BoB coastal regions

also display smaller solar penetration depths, further widening the region impacted by biological surface warming.

The additional biological warming is likely to be non-uniform across the basin and subject to variability during the summer

season. As identified by the observations from the glider and float 631, the SMC contains patches of greater surface chlorophyll

within the main flow and within the eddies and filaments that split off from the SMC. The chlorophyll concentration within





the SMC depends on its strength and location, which affect the entrainment of phytoplankton from the coastal region of Sri
Lanka (Vinayachandran et al., 2004). The SMC strength and location are influenced by the strength of the SLD and the
propagation of Rossby waves from the eastern side of the basin (Webber et al., 2018). Hence, if conditions are conducive for
a strong SMC intercepting the biologically productive coastal regions from June to July, then surface chlorophyll concentration
increases and enhances surface warming. The SLD also fluctuates in strength and position depending on the local wind stress
curl and the propagation of Rossby waves (Webber et al., 2018). Variability in SLD peak strength determines the upwelling
of nutrients to the sunlit layers that sustain high surface chlorophyll concentrations (Thushara et al., 2019). Hence, this would
vary solar penetration depths and periods of enhanced surface warming in the SLD throughout the summer.

The enhanced surface warming during a 15-day break phase in the BSISO, as shown from the $h_2$ = 14 m simulation,
demonstrates the influence that high surface chlorophyll concentrations could have on SST intraseasonal variability (10–30
day time scales). The intraseasonal SST anomalies during the start of the BoBBLE campaign (1–15 July) are ~0.6°C
(Vinayachandran et al., 2018) and previous studies have found the June–July intraseasonal SST variability to be less than 1°C
(Duncan and Han, 2009; Vinayachandran et al., 2012). Our simulations suggest that higher surface chlorophyll (decreasing $h_2$
to 14 m) could generate an SST perturbation equal to ~60% of the intraseasonal SST variability that is observed during the
first half of the BoBBLE campaign. This is a significant modulation of SST and underlines the importance of accounting for
near-surface chlorophyll and its variability in studies of the BSISO.
The modulation of SST by biological warming has important feedbacks to the South Asian monsoon system. Imposing
seasonally varying chlorophyll concentrations in the BoB has been shown to increase rainfall up to 3 mm day$^{-1}$ over Myanmar
during the southwest monsoon onset and over Bangladesh during the autumn intermonsoon (Giddings et al., 2020). However,
little is known about the impact of chlorophyll concentration on the intraseasonal variability of summer monsoon rainfall. The
SST intraseasonal variability is strongly coupled to active and break periods of the BSISO (Fu et al., 2003), and even partly
contributes to the northward and northwestward propagation of convective bands (Gao et al., 2018). The $h_2$ = 14 m simulation
showed increased warming of the ocean surface and hence a more rapid recovery of SST anomalies during the BSISO break
period. This would increase the turbulent heat fluxes to the atmosphere, destabilise the atmospheric boundary layer, and
potentially trigger convection for the following active period sooner. Glider observations in this study have shown that $h_2$
ranges between 10–31 m on submesoscales and that the sub-seasonal temporal variability of $h_2$ strongly depends on the strength
and positioning of the SMC and SLD. Hence, the timing and spatial scale of the chlorophyll blooms in the central BoB relative
to the break periods of the BSISO is an additional factor to consider when modelling intraseasonal convective events.

**Appendix A: Determining in-situ chlorophyll-a concentration from downwelling irradiance**

This Appendix provides a description of the method, key assumptions and quality control process used to derive a proxy
for in situ chlorophyll-a concentration from downwelling irradiance at 490 nm measured by the Argo profiling floats.

Downward irradiance, $E_d$, at wavelength, $\lambda$, decays approximately exponentially as it penetrates through the water column.
The irradiance just below the surface, $E_d(\lambda, -0)$, decays with depth, $z$, at each discretized layer, dz, from the surface, 0, to $z$.



The rate of decay of irradiance, defined as the diffuse attenuation coefficient, $K_d(\lambda, z)$, is allowed to vary in each discretized layer of 1 m thickness. The function is given as:


$$\ln E_d(\lambda, z) = \ln E_d(\lambda, -0) - \sum_1^n K_d(\lambda, z)\, \Delta z \qquad (A1)$$

where $K_d(\lambda, z)$ is defined as the sum of the attenuation of pure seawater ($K_w$) and the attenuation due to biological material ($K_{bio}$). For each discretised layer, $K_w$ is assumed to remain constant, but $K_{bio}$ is allowed to vary in order to derive depth-varying

chlorophyll-a concentration profiles. $K_{bio}$ varies as a non-linear power law function of chlorophyll-a concentration, [Chl-a] (Morel, 1988; Morel and Maritorena, 2001) meaning $K_d(\lambda)$ is defined as:

$$K_d(\lambda) = K_w(\lambda) + \chi(\lambda)\, [\text{Chl-a}]^{e(\lambda)} \qquad (A2)$$

where $\chi(\lambda)$ and $e(\lambda)$ are empirically determined coefficients. Eq. (A1) then becomes,

$$\ln E_d(\lambda, z) = \ln E_d(\lambda, -0) - \sum_1^n \left[ K_w(\lambda) + \chi(\lambda)\, [\text{Chl-a}]^{e(\lambda)} \right] \Delta z. \qquad (A3)$$

Morel et al., (2007) derived the spectrally dependent $\chi(\lambda)$ and $e(\lambda)$ parameters using linear regression analysis of the log-
transformed chlorophyll-a concentration and $K_d$ sourced from the LOV (Laboratoire d'Océanographie de Villefranche) dataset. This study used optical parameters $K_w(490) = 0.0166$, $\chi(\lambda) = 0.08253$  and $e(\lambda) = 0.6529$ for wavelength 490 nm. Measured irradiance values $\ln E_d(490, z)$ and $\ln E_d(490, -0)$ (Fig. A1a) are used to determine $K_{bio}$ for each discretised layer using Eq. (A3) (Fig. A1b and A1c; black dotted line). Further quality control is applied to remove smaller "cloud" spikes (caused by the transient passage of clouds across the sun) from profiles of $K_{bio}$, which is identified as large anomalous alternating positive
and negative spikes. $K_{bio}$ values that are above a threshold of 0.1 m$^{-1}$ and below -0.05 m$^{-1}$ are also removed. The profiles are linearly interpolated onto a 1 m grid and a centered rolling median of window size 15 is applied to smooth over any remaining anomalous noise (Fig. A1c; magenta dotted line). Chlorophyll concentrations are calculated using $\chi(\lambda)$ and $e(\lambda)$ (Fig. A1d; green dotted line).

The relationship between light absorption and chlorophyll-a concentration is assumed to be constant and open ocean water
in the southern BoB is assumed to be categorised as "Case 1" waters, where optical properties are affected by chlorophyll pigments and detrital organic matter (Morel, 1988). The BoB surface ocean mainly consists of chlorophyll-a pigments, as shown from in situ water samples (Madhu et al., 2006), chlorophyll-a fluorescence measurements and remotely sensed satellite measurements (Thushara et al., 2019). Hence, Eq. (A3) and the empirically determined coefficients are suitable to determine chlorophyll-a concentration profiles from $E_d(490)$ measured by the floats.





**Data Availability**

The satellite chlorophyll-a products were produced by the Ocean Colour project European Space Agency Ocean Colour - Climate Change Initiative (ESA OC-CCI; http://www.esa-oceancolour-cci.org) version 3.1 and obtained from the CEDA archive. The absolute dynamic topography and geostrophic velocity products were produced by SSALTO/Duacs, distributed

by AVISO (https://www.aviso.altimetry.fr) and accessed using the Copernicus Marine Environment Monitoring Service (http://marine.copernicus.eu). The Argo profiling float datasets are freely available from the international Argo program (www.argo.ucsd.edu, http://argo.jcommops.org). The glider dataset is available from the British Oceanographic Data Centre (https://www.bodc.ac.uk/data/bodc_database/gliders/).

**Author contribution**

JG performed the data analysis and the KPP simulations. BGMW and MMJ provided support setting up KPP and ASF and BK contributed to Argo float data anaylsis. All co-authors contributed to the design of the study, data interpretation and paper preparation.

**Conflicting Interests**

The authors declare that they have no conflict of interest

**Acknowledgements**

The BoBBLE project is a joint programme funded by the Ministry of Earth Sciences (MOES, Government of India) and the
Natural Environment Research Council (NERC, United Kingdom). The BoBBLE fieldwork campaign was carried out onboard R/V *Sindhu Sadhana*, funded by MOES as part of the Monsoon Mission programme. JG PhD project was supported by NERC EnvEast DTP (NE/L002582/1). NERC supported the fieldwork campaign, BGMW, AJM, KJH and MMJ (NE/L013827/1) and ASF and BK (NE/L013835/1). The authors would like to thank Bastien Queste for use of the UEA glider toolbox to process the glider dataset (http://www.byqueste.com/toolbox.html). The authors would also like to thank Nick Klingaman at
the University of Reading for providing the KPP one-dimensional ocean mixed layer model. The KPP simulations were completed using the High Performance Computing Cluster at the University of East Anglia.

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



| Source | Label | $r^2$ | RMSE [m] |
|---|---|---|---|
| Morel and Antoine (1994) | MA94 | 0.35 | 3.6 |
| Ohlmann (2003) | O03 | 0.35 | 3.7 |


**Table 1: Summary of determination coefficients ($r^2$) and root-mean-square errors (RMSE) when comparing parameterizations to ocean glider (SG579) observed scale depth, $h_2$, and average 0–30 m mixed layer chlorophyll concentrations (Fig. 9).**











**Figure 1: (a) Satellite composite of July 2016 average 4 km chlorophyll-a concentration [mg m⁻³] obtained from ESA OC-CCI version 3.1. The dashed black box shows the outline of Fig. 5a. (b) Absolute dynamic topography [m] of horizontal resolution 0.25° x 0.25° overlaid with surface geostrophic velocity [m s⁻¹] from AVISO for July 2016. (c–f) Satellite composite of 8-daily averaged 4 km chlorophyll-a concentration and surface geostrophic velocities for 1–8 July, 19–27 July, 1–8 August and 17–25 September. Deployment locations and trajectories of glider SG579 (diamond marker; black line), float 629 (square marker; blue line), float 630 (circle marker; green line) and float 631 (triangle marker; red line) are overlaid. Missing data is shaded grey.**


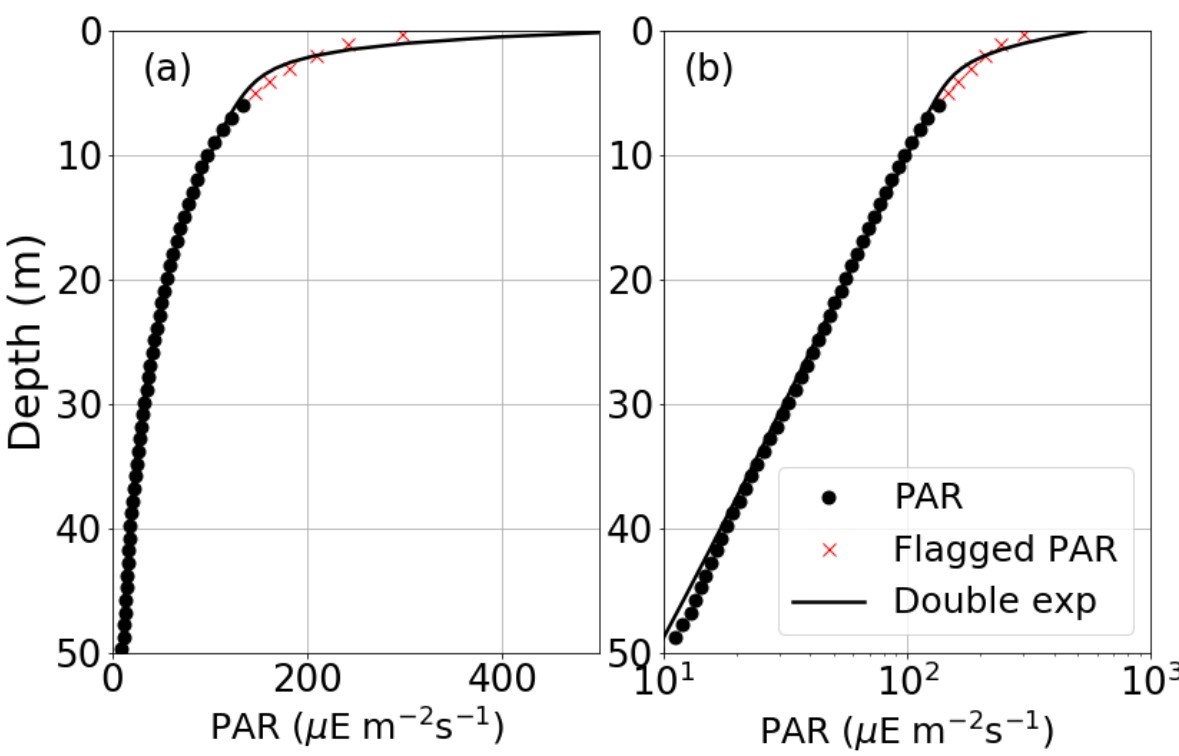

**Figure 2: (a)** Profile of PAR (black circles) measured from float 629 from the surface to 50 m depth with a fitted double exponential function (black line) to PAR between 5–100 m depth. *R* and $h_1$ were specified to be 0.67 and 1.0 m respectively. Red crosses show flagged PAR values that were excluded from the curve fit. **(b)** Same vertical profile of PAR and fitted double exponential function as (a), but presented in log space.






**Figure 3:** Time series of observations measured by glider SG579, linearly interpolated to 1 m depth intervals down to 100 m: (a) temperature [°C], (b) absolute salinity [g kg$^{-1}$], (c) PAR [μE m$^{-2}$ s$^{-1}$], (d) chlorophyll concentration and vertical profile of the average chlorophyll concentration [mg m$^{-3}$]. The black circles are scale depth values, $h_2$ [m]. The mixed layer depth is defined as

the depth where density is same as the surface density plus an increase in density equivalent to a 0.8°C decrease in temperature, and the isothermal layer depth is calculated as the depth where temperature is 0.8°C cooler than SST (Kara et al., 2000; Thushara et al., 2019). The region between the mixed layer depth (grey line) and isothermal layer depth (red line) is the barrier layer.


**Figure 4: (a–d) Time series of observed $h_2$: (a) glider SG579 (black), (b) float 629 (blue), (c) float 630 (green) and (d) float 631 (red). Dashed black lines represents a centered moving average of $h_2$ values with window size of 10 data points. (e–h) Histograms of observed $h_2$ for each glider and floats with the same color scheme as the time series.**



Figure 5: (a) Location of each profile for glider SG579 (diamond), float 629 (square), float 630 (circle) and float 631 (triangle)
across the southern Bay of Bengal colored by the observed $h_2$ value. ESA OC-CCI version 3.1 satellite composite of 4 km
chlorophyll-a concentration for the month of July 2016 is shown. (b) $h_2$ variability with longitude across the basin for glider SG579
(black diamond), float 629 (blue square), float 630 (green circle) and float 631 (red triangle). The grey solid line represents the
mean $h_2$ value binned at 0.5° intervals.






**Figure 6: Time series of observations measured by float 631, linearly interpolated to 1 m depth intervals: (a) temperature [°C], (b) absolute salinity [g kg⁻¹], (c) PAR [µE m⁻² s⁻¹], (d) chlorophyll concentration and vertical profile of the average chlorophyll concentration [mg m⁻³]. Grey sections in the chlorophyll time series represent removed $E_d(490)$ profiles that displayed excessive noise. The black dots are scale depth values, $h_2$ [m]. The grey line for each time series represents the mixed layer depth. The red line represents the isothermal layer depth.**





**Figure 7: As Fig. 6 but for float 629.**





**Figure 8: As Fig. 6 but for float 630.**





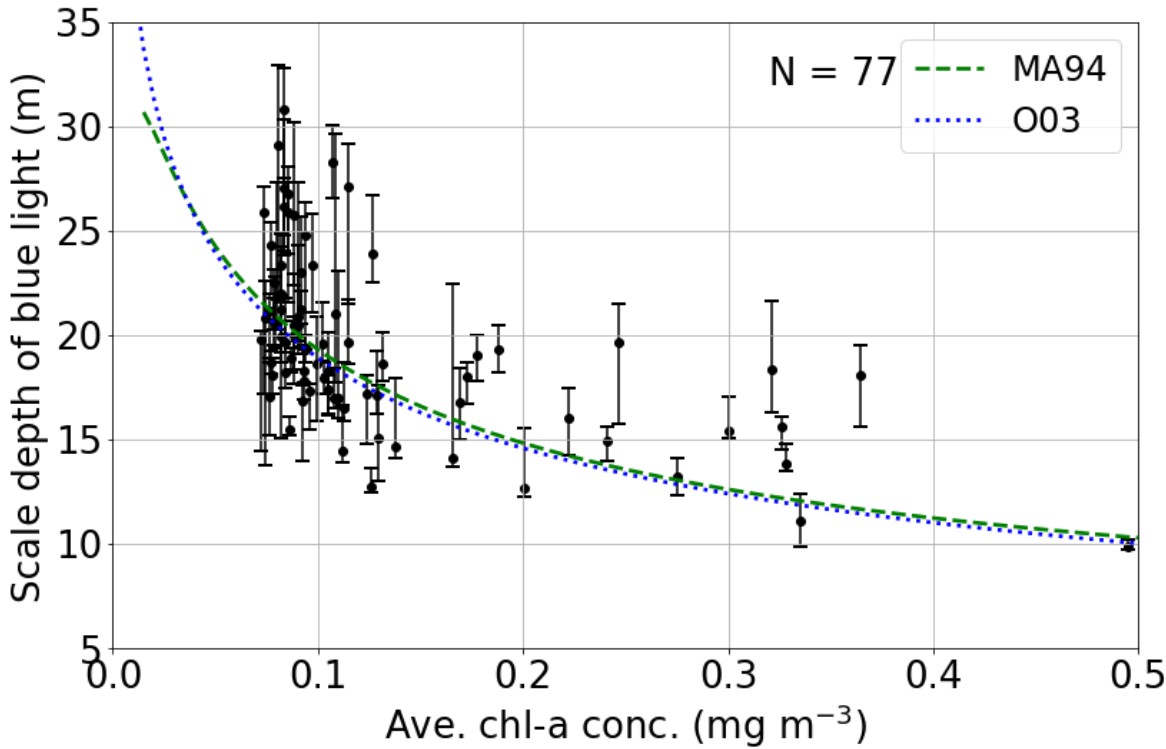

**Figure 9: Observed $h_2$ against average chlorophyll-a concentration between the surface to 30 m depth from glider SG579 (black circles). Parameterisations of scale depth of blue light (equivalent to $h_2$) for chlorophyll concentrations between 0–0.5 mg m$^{-3}$ are presented with the observational data: Morel and Antione (1994) [MA94] (dashed green line) and Ohlmann (2003) [O03] (dotted blue line).**






**Figure 10: (a)** Hourly surface longwave (red line), sensible (green line) and latent (blue line) heat fluxes [W m⁻²] for July 2016; **(b)** Hourly surface shortwave (grey line) and net (black line) heat fluxes [W m⁻²]; **(c)** Wind stress magnitude (dashed black line) [N m⁻²] and precipitation rate (solid black line) [mm day⁻¹]; **(d)** Time series of model SST when $h_2$ is 14 m (black line), 17 m (blue line), 19 m (cyan line), 21 m (green line) and 26 m (red line); **(e)** Time series of daily average SST difference where $SST_{14m}$ minus

$SST_{26m}$ (black line), $SST_{17m}$ minus $SST_{26m}$ (blue line), $SST_{19m}$ minus $SST_{26m}$ (cyan line) and $SST_{21m}$ minus $SST_{26m}$ (green line); **(f)** Time series of model mixed layer depth when $h_2$ is 14 m (black line), 17 m (blue line), 19 m (cyan line), 21 m (green line) and 26 m (red line); **(h)** Depth-time section of salinity [g kg⁻¹] and density (contours) [kg m⁻³] from the $h_2$ = 26 m simulation; **(h)** Depth-time section of temperature difference ($T_{14m} - T_{26m}$) [°C] and density (contours) [kg m⁻³] from the $h_2$ = 26 m simulation.





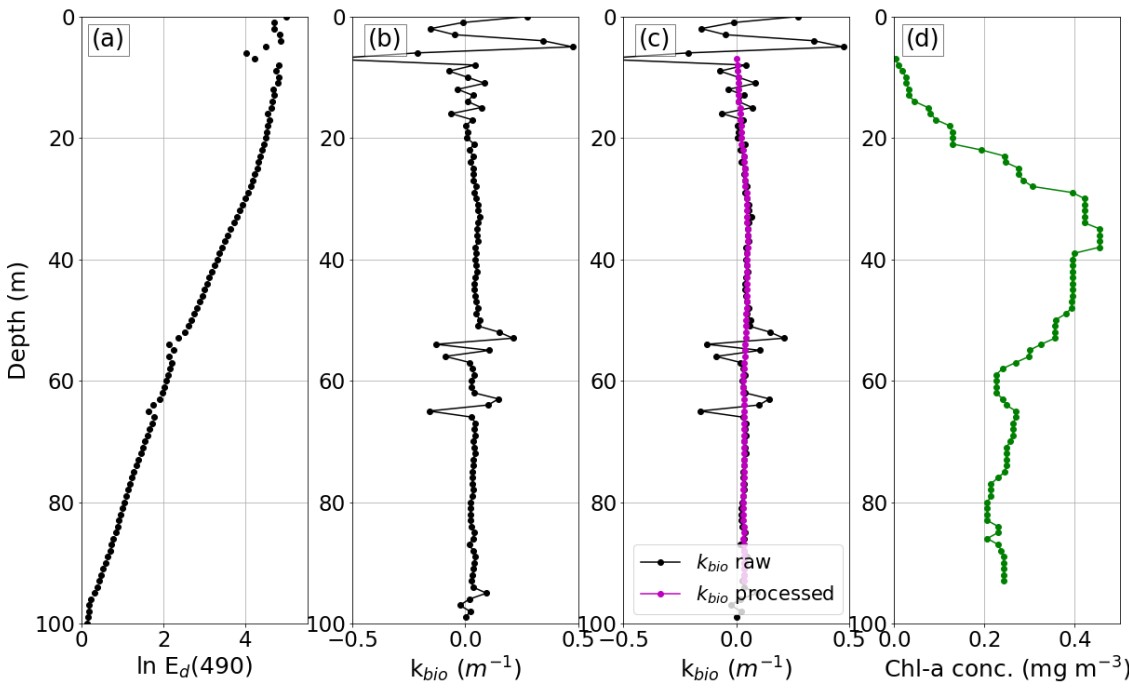


**Figure A1: (a) Profile of ln E$_d$(490) from dive 10, float 631, interpolated onto 1 m vertical grid; (b) The attenuation due to biological constituents, K$_{bio}$ [m$^{-1}$]; (c) K$_{bio}$ [m$^{-1}$] before quality control processing (black dotted line) and after quality control processing (magenta dotted line); (d) Chlorophyll-a concentration [mg m$^{-3}$] (green dotted line).**