# Peer review of "Spatial and temporal variability of solar penetration depths in the Bay of Bengal and its impact on SST during the summer monsoon"

_Ocean Science, 2020_

## Referee Comment (RC1)

**Review**

Spatial and temporal variability of solar penetration depths in the Bay of Bengal and its impact on SST during the summer monsoon

*Authors: Jack Giddings, Karen J. Heywood, Adrian J. Matthews, Manoj M. Joshi, Benjamin G. M. Webber, Alejandra Sanchez-Franks, Brian A. King, and Puthenveettil N. Vinayachandran*

Reviewed by: Isabelle Giddy

General Comments

This study combines observations from floats and gliders together with a simple one-dimensional model to quantify the potential impact that solar radiation absorption by chlorophyll has on Sea Surface Temperature, and consequently precipitation, in the Bay of Bengal. A novel method to derive chlorophyll-a depth profiles from downwelling irradiance is developed and described. The data and results are new and of current scientific interest, warranting publication in Ocean Science.

Below are a number of minor comments which I think could improve the manuscript.

Specific Comments / Technical Corrections (in the order of the manuscript)

1. **paragraph 105:** the scale depth, $h_2$, is not previously defined as $h_2$ in the introduction

2. **paragraph 145**: can it be assumed that the shipboard CTD fluorescence sensor was itself calibrated to in situ bottle samples?

3. **paragraph 220**: are these uncertainties of the scale depth linearly related? And are they quoted later in the text?

4. **paragraph 230:** when averages are quoted it is nice to see standard deviations as well

5. **paragraph 235:** might be nice to be reminded from what values the ML is freshening and warming to, e.g. "….freshens from 34 to 33.3 g kg$^{-1}$"

6. **paragraph 245:** "the variability of h2 is large" (add standard deviation?; Fig 4a)

7. **paragraph 260:** looking at figures 6a and 6b, it appears that the ML only deepens around the 26$^{th}$ July – perhaps mark on the figure the time the period you refer to. Might also be useful to the reader to mark out the barrier layer definition in the caption or on the figure again.

8. **paragraph 260:** "similar to the sub-daily variability of h2 observed from the glider in the SMC." Quote the values or reference figure 3 here

9. **paragraph 265:** Could abbreviate mixed layer depth to MLD here and elsewhere in the text

10. **paragraph 270:**

10.1. The transition from describing the conditions observed by float 629 to that observed by the glider presumably is a bit confusing here. Suggest beginning the following sentence with "in contrast" or "conversely"

Closer to the East India continental shelf, the influence of the freshwater runoff from rivers entering the basin enhances the supply of biological material and the nutrient supply to the upper water column (Lotliker et al., 2016).

9.2 "As a result h2 is reduced" – does this imply h2 getting deeper or shallower. Suggest rephrasing for clarity.

9.3 "Sedimentary material also reduces the solar penetrative depths and increases solar absorption in the surface layers of the coastal region. As a result, h2 is reduced to the west of 83ÅãE (Fig.5b), associated with higher remotely sensed chlorophyll concentrations in this region (Fig. 5a)." – the second sentence here seems to be referring to increased nutrients from river runoff, not sedimentary material.

9.4 …, associated with increased satellite chl-a concentrations. The previous sentence mentions sedimentation also being a factor in setting h2 depth. Suggest relooking at this paragraph for increased readability.

9.5 Add anticyclonic eddy track to supplementary? (maybe not necessary?)

**11. paragraph 280:** "Towards the end of September, the SMC influence at 89° E reduces and the current shifts to the western side of the basin (Fig. 1f), consistent with climatological observations (Webber et al., 2018)."

Suggest changing to active voice: "…at 89E, the influence of the SMC *(on chl-a concentration?)* decreases and the current shifts to the western side …"

**12.** Float 631 yields h2 values greater than 20 m – replace greater with deeper?

**13. Paragraph 290:** The chlorophyll concentration of the surface layer, where the majority of visible radiation is absorbed, is a key control on the amount of visible radiation absorbed and thus on the radiant heating rate of the surface layer. Suggest rewording

**14. Paragraph 340:** "all determined values…" Is this referring to all values of h2 derived from observations during that period?

**15. Paragraph 370:** "from 26 m to 14 m leads to an increase in daily average SST of 0.35°C" suggest "…has the potential to increase daily average SST by 0.35C"

**16.** "Decreasing h2 from 26 m to 17 m, 19 m and 21 m, leads to progressively smaller increases" – this order appears unintuitive. Should one not decrease form 26 m to 21, 19 17? Or perhaps I have misunderstood.

**17. Discussion**
The authors demonstrate that chlorophyll-a concentration impacts the radiative absorption capacity of the surface ocean. While shallower scale depths induce larger changes in SSTs, it appears that the net impact of this warming is dependent on the depth of the mixed layer – which itself has multiple forcing mechanisms. Particularly, there is a large body of literature which discusses submesoscale variability which could be mentioned in the discussion on implications and assumptions. The assumption that the region is 1D forced should be discussed given the available literature on submesoscale 3D processes active in the BoB. It could also be interesting to suggest possible links between horizontal processes of SMS, shoaling of ML/added nutrients and

the link the chl-a concentration and warmer waters. Suggested literature: Ramachandran et al., 2018; Jaeger and Mahadevan, 2018; Shroyer et al., 2020.

18. **General:** punctuate equations

19. Average chl-a in surface 0-30 m is repeated a number of times throughout the text. Suggest defining and abbreviating at the beginning.

20. **Figure 10:** It is difficult to see differences between simulations in figure 10f. Suggest zoomed in inset.

21. A1d – would it be worth plotting a chl-a profile from the glider compared the float 629 which look to be close in space/time (looking at Figure 5?)

---

## Author Comment (AC1)

**Author Comments 1 on "Spatial and temporal variability of solar penetration depths in the Bay of Bengal and its impact on SST during the summer monsoon"**

We would like to thank Reviewer 1 who provided constructive comments and interesting questions that have improved the revised manuscript. Reviewer 1's comments have been reproduced in black with the authors response in blue and excerpts from the revised manuscript in italics. The revised and renumbered figures are included at the end of the document.

**Response to Reviewer 1**

**Specific comments and technical corrections**

1.  Paragraph 105: the scale depth, $h_2$, is not previously defined as $h_2$ in the introduction.
    Thank you for spotting this. We have now introduced the scale depth ($h_2$) in the preceding paragraph.

    Line 107: *"...allowing us to determine the downward penetration of solar radiation, as represented by the length scale associated with the absorption of blue light, which is represented by the parameter $h_2$."*

2.  Paragraph 145: can it be assumed that the shipboard CTD fluorescence sensor was itself calibrated to in situ bottle samples?
    The shipboard CTD fluorescence sensor was not calibrated to in situ bottle samples, and so is not mentioned in the methodology.

3.  Paragraph 220: are these uncertainties of the scale depth linearly related? And are they quoted later in the text?
    The individual source uncertainties that produce the overall uncertainty in $h_2$ are not linearly related. They are shown as error bars on Figs. 3 to 9 but are not quoted in the text. We have now added the uncertainties of $h_2$ throughout the text in Section 3. We have better explained the method of $h_2$ uncertainty.

    Line 228: *"We combine the maximum and minimum values of each source of uncertainty to calculate the upper and lower uncertainty bounds of each derived value of $h_2$."*

4.  Paragraph 230: when averages are quoted it is nice to see standard deviations as well.
    The Reviewer is correct in suggesting that standard deviations should be supplied with average values. These have now been added throughout Section 3.1.

5.  Paragraph 235: might be nice to be reminded from what values the ML is freshening and warming to, e.g. "....freshens from 34 to 33.3 g kg$^{-1}$".
    Thank you for your suggestion, we have reminded the reader what values the ML has freshened and warmed from and included standard deviations from the previous comment.

    Line 244: *"Within the SMC, the mixed layer warms from 28.0 to 29.0 ± 0.2 °C and freshens from 34.0 to 33.3 ± 0.1 g kg$^{-1}$ (Fig. 3a and 3b)."*

6.  Paragraph 245: "the variability of h2 is large" (add standard deviation?; Fig 4a).
    The standard deviation has now been added to give greater clarity.

Line 252: *"The temporal variability of $h_2$ in the SMC is large with a standard deviation of 4 m (Fig. 4a)."*

7. Paragraph 260: looking at figures 6a and 6b, it appears that the ML only deepens around the 26[th] July – perhaps mark on the figure the time the period you refer to. Might also be useful to the reader to mark out the barrier layer definition in the caption or on the figure again.
We have highlighted the section of the Figure we refer to in the text using solid black lines. This is now referenced in the text. The definition of the barrier layer is repeated for the reader in Figure 6 caption. Please see revised Figure and caption at the end of the document.

Line 268: "…as barrier layer thickness increases to 40 m (*area between two solid black lines*; Fig. 6a and 6b)."

8. Paragraph 260: "similar to the sub-daily variability of h2 observed from the glider in the SMC." Quote the values or reference figure 3 here.
We have reminded the reader what the variation in $h_2$ is for the glider.

Line 270: *"Average values of $h_2$ are around $16 \pm 1$ m, varying between 10 to 20 m, smaller than the 15 to 31 m sub-daily variability of $h_2$ observed from the glider in the SMC."*

9. Paragraph 265: Could abbreviate mixed layer depth to MLD here and elsewhere in the text.
We have now defined the acronym on first encounter (line 41) and thereafter abbreviated instances of mixed layer depth to MLD throughout the manuscript.

10. Paragraph 270:

10.1. The transition from describing the conditions observed by float 629 to that observed by the glider presumably is a bit confusing here. Suggest beginning the following sentence with "in contrast" or "conversely".
We agree with Reviewer 1 and have now improved the readability of this paragraph.

Line 272: *"Conversely, observations on the western side of the basin from float 629, between 8 and 11° N, show average $h_2$ values of 20 m compared with the average $h_2$ values of 16 m in the SMC from SG579 (Fig. 5a)."*

10.2 "As a result h2 is reduced" – does this imply h2 getting deeper or shallower. Suggest rephrasing for clarity.
$h_2$ is defined as a length scale, so it increases and decreases, not deepens or shoals.

10.3 "Sedimentary material also reduces the solar penetrative depths and increases solar absorption in the surface layers of the coastal region. As a result, h2 is reduced to the west of 83E (Fig.5b), associated with higher remotely sensed chlorophyll concentrations in this region (Fig. 5a)." – the second sentence here seems to be referring to increased nutrients from river runoff, not sedimentary material.
Thank you for pointing this out, we are sorry for the confusion. We have now clarified the last sentence.

Line 283: *"As float 629 approaches the East India continental shelf, $h_2$ is reduced to the west of 83°E (Fig. 5b), likely due to high chlorophyll concentrations and sedimentary material in this region as captured by satellite (Fig. 5a)."*

10.4 ..., associated with increased satellite chl-a concentrations. The previous sentence mentions sedimentation also being a factor in setting h2 depth. Suggest relooking at this paragraph for increased readability.

As mentioned in our response to 10.3 we have improved the readability of this paragraph.

10.5  Add anticyclonic eddy track to supplementary? (maybe not necessary?)
This is an interesting suggestion, but we argue that adding the track of the anticyclonic eddy to the supplementary section is not necessary and may distract the reader.

11. Paragraph 280: "Towards the end of September, the SMC influence at 89° E reduces and the current shifts to the western side of the basin (Fig. 1f), consistent with climatological observations (Webber et al., 2018)."  Suggest changing to active voice: "...at 89E, the influence of the SMC *(on chl-a concentration?)* decreases and the current shifts to the western side ..."
    We agree with the Reviewer's suggestion and have now changed the sentence.

    Line 288: *"Towards the end of September at 89° E, the influence of the SMC on chlorophyll concentration decreases as the SMC shifts to the western side of the basin away from float 630 (Fig. 1f), consistent with climatological observations (Webber et al., 2018)."*

12. Float 631 yields h2 values greater than 20 m – replace greater with deeper?
    As in comment 10.2, $h_2$ is defined as a length scale, and so we respectfully disagree with the Reviewer's suggestion here.

13. Paragraph 290: "The chlorophyll concentration of the surface layer, where the majority of visible radiation is absorbed, is a key control on the amount of visible radiation absorbed and thus on the radiant heating rate of the surface layer." Suggest rewording.
    Thank you for your suggestion. We have now reworded this sentence.

    Line 299: *"The majority of visible radiation is absorbed at the near surface, hence the chlorophyll concentration at the near surface strongly influences the amount of visible radiation absorbed, which strongly influences the radiant heating rate of the ocean surface."*

14. Paragraph 340: "all determined values..." Is this referring to all values of h2 derived from observations during that period?
    Reviewer 1 is correct in that all determined values from the glider and floats were used to calculate the $h_2$ percentiles. We have clarified this in the text.

    Line 349: *"... all determined values of $h_2$ from the glider and floats respectively throughout July 2016."*

15. Paragraph 370: "from 26 m to 14 m leads to an increase in daily average SST of 0.35°C" suggest "...has the potential to increase daily average SST by 0.35C".
    We have changed the wording of this specific statement to better reflect what happened in our experiment.

    Line 378: *"In the idealised KPP experiments, changing $h_2$ from 26 m to 14 m led to an increase in daily average SST by 0.35°C within a month (black line; Fig. 10e)."*

16. "Decreasing h2 from 26 m to 17 m, 19 m and 21 m, leads to progressively smaller increases" – this order appears unintuitive. Should one not decrease form 26 m to 21, 19 17? Or perhaps I have misunderstood.
    We agree with Reviewer 1 in that it would be more intuitive if the order is the other way round.

    Line 381: *"Decreasing $h_2$ from 26 m to 21 m, 19 m and 17 m, leads to progressively larger increases in daily average SST from 0.14°C, 0.18°C and 0.25°C by the end of July 2016, respectively (Fig. 10e)."*

17. Discussion: The authors demonstrate that chlorophyll-a concentration impacts the radiative absorption capacity of the surface ocean. While shallower scale depths induce larger changes in SSTs, it appears that the net impact of this warming is dependent on the depth of the mixed layer – which itself has multiple forcing mechanisms. Particularly, there is a large body of literature which discusses submesoscale variability which could be mentioned in the discussion on implications and assumptions. The assumption that the region is 1D forced should be discussed given the available literature on submesoscale 3D processes active in the BoB. It could also be interesting to suggest possible links between horizontal processes of SMS, shoaling of ML/added nutrients and the link the chl-a concentration and warmer waters. Suggested literature: Ramachandran et al., 2018; Jaeger and Mahadevan, 2018; Shroyer et al., 2020.
We have added an additional paragraph to the discussion outlining the limitations to KPP.

> Line 458: *"KPP is a one deimensional model and neglects horizontal advection. Submesoscale frontal and eddy activity in the BoB create sharp horizontal and vertical gradients in temperature and salinity (Ramachandran et al., 2018; Jaeger and Mahadevan, 2018). Strong seasonal surface currents, such as the SMC, advect different water masses, forming fronts and eddies that are continually moving and changing around the BoB. This submesoscale dynamical variability is not replicated in the one dimensional KPP model. However, for the purposes of this paper, the simplicity inherent in not representing three dimensional dynamics means that the results of our chlorophyll sensitivity experiments are unambiguous."*

18. General: punctuate equations.
Thank you for spotting this. We have now punctuated the equations.

> E.g. Line 485: "$\ln E_d(\lambda, z) = \ln E_d(\lambda, -0) - \sum_1^n K_d(\lambda, z) \, \Delta z$,"

19. Average chl-a in surface 0-30 m is repeated a number of times throughout the text. Suggest defining and abbreviating at the beginning.
Thank you for your suggestion. We have now abbreviated 0-30 m chlorophyll concentration to $Chl\text{-}a_{30}$. This has been defined in Section 3.1.

> Line 240: *"... (henceforth referred to as Chl-a$_{30}$) ..."*

20. Figure 10: It is difficult to see differences between simulations in figure 10f. Suggest zoomed in inset.
Thank you for your suggestion. We have provided a zoomed in inset of the MLD time series for the period of 24 to 30 July, which is labelled as Fig. 10g. We have referenced this new panel in the text.

21. A1d – would it be worth plotting a chl-a profile from the glider compared the float 629 which look to be close in space/time (looking at Figure 5?)
As mentioned in our comment to Reviewer 2, the distance between the deployment location of float 629 and SG579 is approximately 56 km, which is large enough for noticeable changes in chlorophyll concentration. Hence, we have not added the chlorophyll concentration profiles in the same plot as it would not provide a useful comparison with fluorescence-derived chlorophyll concentration.

**References**

Jaeger, G. S. and Mahadevan, A.: Submesoscale-selective compensation of fronts in a salinity-stratified ocean, Science Advances, 4, 1701504, https://advances.sciencemag.org/lookup/doi/10.1126/sciadv.1701504, 2018.

Ramachandran, S., Tandon, A., Mackinnon, J., Lucas, A. J., Pinkel, R., Waterhouse, A. F., Nash, J., Shroyer, E., Mahadevan, A., Weller, R. A. and Farrar, J. T.: Submesoscale processes at shallow salinity fronts in the Bay of Bengal: Observations during the winter monsoon, J. Phys. Oceanogr., 48, 479–509, https://journals.ametsoc.org/view/journals/phoc/48/3/jpo-d-16-0283.1.xml, 2018.

[Figure]

**Figure 6: Time series of observations measured by float 631, linearly interpolated to 1 m depth intervals: (a) temperature [°C], (b) absolute salinity [g kg⁻¹], (c) PAR [µE m⁻² s⁻¹], (d) chlorophyll concentration and vertical profile of the average chlorophyll concentration [mg m-3]. Grey sections in the chlorophyll time series represent removed E$_d$(490) profiles that displayed excessive noise. The black dots are scale depth values, h$_2$ [m]. The region between the MLD (grey line) and isothermal layer depth (red line) is the barrier layer.**

[Figure]

**Figure 10: (a)** Hourly surface longwave (red line), sensible (green line) and latent (blue line) heat fluxes [W m⁻²] for July 2016; **(b)** Hourly surface shortwave (grey line) and net (black line) heat fluxes [W m⁻²]; **(c)** Wind stress magnitude (dashed black line) [N m⁻²] and precipitation rate (solid black line) [mm day⁻¹]; **(d)** Time series of model SST when $h_2$ is 14 m (black line), 17 m (blue line), 19 m (cyan line), 21 m (green line) and 26 m (red line); **(e)** Time series of daily average SST difference where $SST_{14m}$ minus $SST_{26m}$ (black line), $SST_{17m}$ minus $SST_{26m}$ (blue line), $SST_{19m}$ minus $SST_{26m}$ (cyan line) and $SST_{21m}$ minus $SST_{26m}$ (green line); **(f)** Time series of model mixed layer depth when $h_2$ is 14 m (black line), 17 m (blue line), 19 m (cyan line), 21 m (green line) and 26 m (red line); **(g) Time series of model mixed layer depth between 24 and 30 July;** **(h)** Depth-time section of salinity [g kg⁻¹] and density (contours) [kg m⁻³] from the $h_2 = 26$ m simulation; **(i)** Depth-time section of temperature difference ($T_{14m} − T_{26m}$) [°C] and density (contours) [kg m⁻³] from the $h_2 = 26$ m simulation.

---

## Author Comment (AC2)

**Author Comments 2 on "Spatial and temporal variability of solar penetration depths in the Bay of Bengal and its impact on SST during the summer monsoon"**

We would like to thank Reviewer 2 who provided constructive comments and interesting questions that have improved the revised manuscript. Reviewer 2 comments have been reproduced in black with the authors response in blue and excerpts from the revised manuscript in italics.

**Response to Reviewer 2**

**Specific comments:**

1. Section 3.1: The glider measurements are discussed to explain the Chl-a variations in time vs. depth over the region of glider deployment. The BoB is known for having sharp horizontal gradients of properties (T, S, and maybe Chl-a). In the eddy region, these sharp gradients are likely to form. However, the results discussed in this section appear to assuming spatial homogeneity in the area covered by the glider trajectory. One possible solution could be to plot along-track profiles.

The reviewer is correct that we focus our discussion of spatial variations on the more widely spaced float data, and focus our discussion of temporal variability on the glider, which was occupying a time series site in virtual mooring mode. Along track profiles are not helpful since the glider spent the majority of it's time at one location. Of course any time series location is a mixture of temporal and spatial variability as eddies and fronts are advected past the glider. Nonetheless the KPP modelling in section 3.3 confirms the value of treating the glider data as a time series. We now make this clearer in the text.

> Line 243: *"Patches of surface chlorophyll, with concentrations of 0.1–0.4 mg m$^{-3}$ (Fig. 1d), continue to be advected by the SMC into the region where glider SG579 is parked at a virtual mooring at 85° E until 19 July."*

> Line 252: *"The temporal variability of $h_2$ in the SMC is large with a standard deviation of 4 m (Fig. 4a)."*

2. In Figure 2, What are causes of the measured (flagged) PAR values departing from the fit. Do we consider PAR values inaccurate in upper few meters or the double exponential fit method is not well-suitable close to surface? How would it affect the calculation of heat terms and SST (change in SST due to Chl-a)?

We would like to thank the Reviewer for their interesting questions. Flagged PAR values in the top 5 m depart from the fit due to (i) noise caused by wave-focusing and cloud shadows, and (ii) the double exponential function is not well suited close to the surface. This has now been added to Section 2.2. As mentioned in Section 2.2, the two-band model is only an approximation of the decay of visible radiation with depth across the full solar spectrum. Increasing the number of exponential terms would improve the fit to near-surface PAR, as the number of degrees of freedom increases. However, for the purposes of this study, we use the two-band model as it is commonly used in coupled ocean-atmosphere GCMs and we do not have confidence in fitting a higher band model to near-surface PAR as we would likely be fitting to noise. Due to the GCMs coarse vertical resolutions, all long wavelengths are absorbed within the first layer of a model ocean. The use of a higher band model would have little influence on the radiant heating rate and SST in these GCMs.

> Line 224: *"Generally, flagged PAR values in the top 5 m depart from the fit due to excessive noise caused by wave-focusing and cloud shadows, and the poor approximation of Eq. (1) representing the absorption of longer wavelengths near the surface."*

3. The glider and float 629 are very close in the first week of July (as seen in Fig 1) but their h2 values differ a lot and appears out of phase between these two measurements. Is it due to different sensors used on glider and float or a calibration issue, or due to any other process?

The distance between the deployment location of float 629 and SG579 is approximately 56 km, which is large enough for noticeable differences in chlorophyll concentration and $h_2$ values. SG579 is deployed in the middle of the SLD, whilst float 629 is deployed on the western side of the SLD. Conditions inside the SLD are conducive for increased biological activity, hence SG579 observes higher chlorophyll concentrations and correspondingly smaller $h_2$ values compared with float 629 on the outer edges of the SLD. Although the sensors are different on the glider and float, this has a minimal effect on final $h_2$ values. As mentioned in Section 2.2, line 215, the absorption rate of PAR with depth is independent of the absolute values of PAR.

4. Line 369-293: The effect of changing h2 depths on the SST is described here. A major concern is that the SST differences among different h2 values prescribed in model shows a progressively increasing differences in SST with the increasing time of simulation. At the beginning of simulation all the SST curves are aligned and by the end of July month, the difference is largest. This points to the possible issue with a drift in model. The precipitation events after 15th July changes the absolute magnitude of SST in all experiments but the difference in SST remains unaffected by precipitation.

We respectfully disagree with the reviewer. We observe no signs of drift in the KPP model when we initially 'spin-up' the KPP model for the month of June 2016. Differences in SST are caused by changes in $h_2$, or the absorption rate of blue light with depth, as determined from PAR measurements from SG579. These differences are due to different net heat fluxes that accumulate over time to cause the divergence noted by the reviewer.

5. Lines 63-74: Assimilation of satellite-derived Chlorophyll (Chl) concentration would improve the simulation of Chl on the surface. But the radiation attenuation occurs in the water column. How these climate models simulate the vertical profiles of the Chl? That would determine their ability to correctly representing radiation attenuation in the mixed layer and, therefore, the SST simulation.

Thank you for your question. Generally, ocean GCMs do not explicitly model the vertical profiles of chlorophyll concentrations because of the computational expense. Instead, satellite-derived chlorophyll concentration values are assigned to each ocean grid point, representing the average chlorophyll concentration between the surface and one scale depth. Using the chlorophyll-dependent parameterisations, the satellite-derived chlorophyll concentrations are converted into a solar penetration depth, which can then be used in the solar radiation scheme of the ocean model. This has now been better explained in Section 1.

> Line 69: *"Both GCMs have the capability to assimilate satellite-derived chlorophyll concentrations. These chlorophyll concentrations can then be converted into a solar penetration depth using chlorophyll-dependent parameterisations. Satellite-derived chlorophyll concentrations have revolutionized our understanding of how chlorophyll-induced heating affects ocean dynamics and the climate system (Murtugudde et al., 2002; Sweeney et al., 2005; Wetzel et al., 2006)."*

6. Lines 196-201: Radiation penetration is wavelength dependent and its attenuation is a function of Chl-a concentration and water quality. Red wavelengths are absorbed in top 1-2 m but there are other intermediate wavelengths between red and blue. Only red and blue wavelength bands are referred. What

happens to other intermediate wavelengths? How these are treated? Would it affect the overall estimate of SST change?

We do refer to the two-band model as "red" and "blue" light, but these two bands together encompass all wavelengths of visible light. As mentioned in Section 2.2, line 206, The two-band model is the simplest approximation of the full solar spectrum.

7. The water types are determined dynamically in space and time? Can we consider water type (h2 value) to be same for a period of one month?

The reviewer is correct to point out that the value of $h_2$ may vary temporally as the plankton blooms wax and wane. However, the point of our idealised modelling experiments is to demonstrate quantitatively the impact of different values of $h_2$, rather than to simulate exactly the observed variations; that would require a much more complex numerical model. We now make these caveats more explicit by including the word 'idealised' throughout Section 3.3.

8. Line 241: 'The position and velocity of the SMC relative…..south-central BoB'. There could be some contribution to the Chl-a in the south-central BoB from the productive southwest coast of India (apart from the source in the south of Sri Lanka).

The Reviewer is correct in suggesting that biologically productive water along the southwest coast of India also contributes to the chlorophyll concentrations in the south-central BoB. The following sentence has been edited:

> Line 248: *"The position and velocity of the SMC relative to the biologically productive southern coast of Sri Lanka and southwest coast of India determines how much surface chlorophyll is entrained and advected into the south-central BoB (Vinayachandran et al., 2004)."*

9. In Figure 3(d), Chl-a increases in near-surface layers during 16-17 July. What are possible reasons for this increase? Is it advection-driven due to a chance of upwelling (noticing a decrease in Chl-a just below the thermocline in the corresponding period).

The increase in chlorophyll concentration in the top 30 m during 16-17 July is likely due to horizontal advection of fresher and more biologically productive water. Upwelling is unlikely as Fig. 1d shows that the SLD has weakened by the end of July and there is no indication that the thermocline is doming, leading to upwelling and increased salinity. The vertical distribution of chlorophyll concentration is affected by additional factors that are not investigated in this study, such as nutrient supply, light limitation, grazing, mortality and sinking rates (Thushara et al., 2019). Thus, it is difficult to identify the direct cause(s) of increased chlorophyll concentration.

10. Line 360: Apart from the varying h2 values (14 m, 17 m, 19 m, 21 m and 26 m), you also have changing R values in different experiments? Since the two parameters are being changed in each sensitivity expt, one should be careful in checking that it should not affect the inferences drawn from the experiments (i.e. relating to only h2 variations).

The Reviewer is correct in that we vary the value of R and $h_2$ in our idealised KPP simulations. Sensitivity experiments, not in the present study, have shown that variations in $h_2$ contribute to the largest changes in SST, whereas variations in R contribute to small, non-negligible changes in SST. Therefore, we conclude that it is primarily the variations in $h_2$ that are responsible for the SST changes in our idealised experiments.

> Line 351: *"Initial idealised KPP sensitivity experiments, not presented in this paper, show that the influence of R on SST is not negligible but the influence of $h_2$ on SST is the largest out of all optical parameters."*

11. In Abstract: Chlorophyll influences regional climate through its effect on solar radiation absorption and thus sea surface temperature (SST) --- Chlorophyll affects climate through other processes as well (e.g. air-sea gas exchange, CO2 uptake).

The Reviewer is correct in that chlorophyll does affect air-sea gas exchange such as CO2 drawdown. This has now been highlighted in the Abstract.

> Line 13: *"Chlorophyll has long been known to influence air-sea gas exchange and $CO_2$ drawdown. But chlorophyll also influences regional climate through its effect on solar radiation absorption and thus sea surface temperature (SST)."*

12. Mention in figure caption- what do the error bars indicate in figure 4?

The error bars indicate the uncertainty of derived values of $h_2$. We have clarified this by adding an additional sentence to the figure caption.

> Fig. 4 caption: *"The error bars indicate the uncertainty of derived values of $h_2$."*